# Performance Evaluation of the New Chemiluminescence Immunoassay CL-1200i for HBV, HIV Panels

**DOI:** 10.3390/diseases11020083

**Published:** 2023-06-08

**Authors:** Eleonora Nicolai, Serena Sarubbi, Martina Pelagalli, Valerio Basile, Alessandro Terrinoni, Marilena Minieri, Oreste Cennamo, Sandro Grelli, Sergio Bernardini, Massimo Pieri

**Affiliations:** 1Department of Experimental Medicine, University of Rome Tor Vergata, Via Montpellier 1, 00133 Rome, Italy; nicolai@med.uniroma2.it (E.N.); serena.sarubbi@gmail.com (S.S.); martina90c@hotmail.it (M.P.); vale.basi90@hotmail.it (V.B.); alessandro.terrinoni@uniroma2.it (A.T.); minieri@med.uniroma2.it (M.M.); grelli@med.uniroma2.it (S.G.); sergio.bernardini@ptvonline.it (S.B.); 2Department of Laboratory Medicine, Tor Vergata University Hospital, Viale Oxford 81, 00133 Rome, Italy; orestecennamo92@gmail.com

**Keywords:** HIV detection, HBV detection, chemiluminescence immunoassay

## Abstract

Infectious diseases such as HIV and HBV are a global concern for their impact in terms of public health and costs for national health services. A central role in contrasting the spread of the infections is represented by timely diagnosis. The speed of detection depends on several factors including the type of test used. Antibody response to hepatitis B surface antigens (anti-HBs) is an important serological marker used for HBV-infection detection. The aim of this study was to compare the performance of the Abbott system and of the new analyser Mindray 1200i in the detection of HBV- and HIV-infections. Clinical serum samples were collected from patients randomly selected from PTV University Hospital of University of Rome “Tor Vergata” and tested for HBV and HIV antibodies. Samples were evaluated by Mindray Cl 1200i CLIA screening tests for HBV and HIV and the results were compared with the Abbott Architect analytical system, the routine instrument of the hospital clinical biochemistry laboratory. Precision study, linearity, and carryover were performed on the results obtained. The agreement between the results of the Abbott and Mindray CLIA ranged from 99% to 100% and the discrepancy rate from 0% to 1%. The measurements demonstrated that the Mindray CL-1200i platform offers high-level performance with accurate and consistent test results and could represent a valuable tool if implemented in routine analysis.

## 1. Introduction

In the global context, the spread and consequences of infectious diseases such as Human immunodeficiency virus (HIV), Hepatitis B virus (HBV), Hepatitis C virus (HCV), and syphilis have significant implications for health, quality of life, and societal costs, both for individuals and the community. These diseases are closely associated with individual behaviors, and the risk is significantly heightened by addiction phenomena [1]. Over the past few decades, our country, as well as Europe, have implemented significant preventive measures against these infectious diseases, yielding appreciable results. However, despite the ongoing efforts to prevent primary infectious diseases, the proportion of individuals tested for these diseases within the general population in hospital care settings remains low. Consequently, it is crucial to enhance screening activities among individuals utilizing addiction services through effective procedures that encompass a broader range of information dissemination, awareness campaigns, and targeted preventive actions aimed at the general population [2,3].

HIV infection does not exhibit specific manifestation but rather manifests through its effects on the immune system. Individuals can live for years without experiencing symptoms and may only become aware of the infection when opportunistic infections arise. Therefore, HIV testing is the only means of detecting the infection. Scientific evidence highlights the positive impact of early therapy on prognosis and quality of life. The incidence of new HIV diagnoses experienced a slight decrease between 2012 and 2015, with a relatively stable trend thereafter. The incidence trends in recent years have been similar across all modes of transmission. In 2017, the highest incidence of HIV infection was observed in the 25–29 age group with heterosexual intercourse being the primary mode of transmission among new HIV diagnoses [4]. HIV infection can be detected through specific blood tests that identify the presence of antibodies and the p24 viral antigen. This is the only way to detect infection; unfortunately, no test can detect the virus immediately after infection. The time required to detect the infection depends on various factors, including the type of test employed [5]. Various types of blood tests, commonly used for HIV diagnosis, search for specific antibodies in the blood and provide results after different periods of time from the last risk behavior. For instance, combined tests (fourth generation tests) detect both anti-HIV antibodies produced by the organism and viral proteins such as the p24 antigen [6]. These tests can identify the infection as early as 20 days, while the window period extends to 40 days from the last risk behavior. Another type of test exclusively looks for anti-HIV antibodies (third generation test) and it can detect the infection approximately 3–4 weeks after the last risk behavior [7].

HBV is a DNA virus from the Hepadnaviridae family. The virus exhibits notable resistance outside the human body. Transmission typically occurs through parenteral means, through needles or surgical instruments, sexual contact, or during pregnancy [8]. The incubation period for the disease is relatively long, ranging from 45 to 180 days. Initial symptoms, such as fatigue, fever, nausea, vomiting, and abdominal pain, precede the onset of icterus (yellowing of the skin and eyes) accompanied by dark urine. Chronic HBV infection affects more than 275 million individuals worldwide, leading to 0.8 million deaths annually, yet only 10% have been diagnosed [9]. Chronic HBV infection presents a significant global health concern, with over 350 million individuals affected. Approximately one-third of the world’s population demonstrates serologic evidence of exposure, underscoring the high burden of disease [10]. In May 2016, the United Nations Sustainable Development Goals established the aim of eliminating HBV infection by 2030, endorsing the Global Health Sector Strategy (GHSS) on viral hepatitis 2016–2021. The GHSS outlines the goal of eliminating viral hepatitis as a public health threat by 2030 (reducing new infections by 90% and mortality by 65%) [11]. Access to affordable hepatitis testing is limited, resulting in a low number of diagnosed cases among individuals with viral hepatitis (only 9% of HBV-infected persons). Moreover, of those diagnosed with HBV infection, only a mere 8% (equivalent to 1.7 million people) are receiving treatment [11]. Currently, several diagnostic assays are available for quantifying HBsAg, and chemiluminescence has garnered increasing attention in clinical diagnosis due to its high sensitivity, good specificity, simple equipment requirements, and wide linear range. Additionally, the antibody response to anti-HBs is an important serological marker for sero-conversion or sero-protection resulting from natural infection or vaccination [12]. Worldwide, hepatitis B virus infection is one of the most common coinfections with human immunodeficiency virus and has become a major threat to the survival of HIV-infected persons. Reducing the disease burden for HIV and HBV requires screening campaigns targeting the ‘silent’ infections and bringing the infected individuals to treatment. The implementation of screening methods for patients with a faster linkage to care for infected individuals has been demonstrated both effective and sustainable. To achieve this, the following is mandatory: update our knowledge on the diseases burden from HIV and HBV and eventually HCV; understand the current hurdles and limitations in finding ‘silent’ infections; recognize the operative and clinical advantages of new screening methods; and highlight the contribution of the clinical laboratory on the viral elimination programs. In line with this clinical requirement, the aim of this study is to compare the performance of the Abbott system and the new Mindray 1200i analyser in detecting HBV and HIV infections. Serum samples collected from patients at the “Tor Vergata” University Hospital of Rome were subjected to testing for various parameters, including the detection of antibodies to the core antigen (Anti HbC), the surface anti-antigen antibody (Anti HbS), the antigen “e” virus antibody (Anti HbE), the antigen “e” of the virus (HbE Ag), the surface HBV antigen (HbS Ag), the HIV p24 antigen, and antibodies to HIV-1 and/or HIV-2.

## 2. Materials and Methods

### 2.1. Patient Samples

This study includes serum samples from randomly selected patients, including both hospitalized and ambulatory patients from PTV University Hospital of University of Rome “Tor Vergata”, in the period June 2021–December 2022. A total of 541 clinical serum samples were collected and tested for HBV and 181 for HIV.

Samples were evaluated by Mindray Cl 1200i CLIA screening tests for HBV and HIV and the results were compared with Abbott Architect analytical system, the routine instrument of the hospital’s clinical biochemistry laboratory. 

Samples were collected in BD Vacutainers tube with clot activator and separating gel (Becton Dickinson UK), processed according to manufacturer specification, and analysed within four hours from veni puncture.

Serum samples were tested immediately after collection and pre-analytical treatment, then centrifuged at 2000 g for 10 min to allow for clot formation. A single test requires 50 μL of sample volume. All tests were performed and interpreted in accordance with the manufacturers.

This study was approved by the hospital ethics committee and conducted according to the revised declaration of Helsinki. (R.S.202/19).

### 2.2. Assay Description

The Mindray CL-1200i is a fully automated chemiluminescence immunoassay (CLIA) system designed for high-throughput testing in clinical laboratories and offers a wide range of immunoassays, including infectious disease assays such as the HIV and HBV assay. The assay follows the principle of the chemiluminescent immunoassay, where specific antigens or antibodies in the patient’s blood sample react with corresponding labeled reagents. The resulting chemiluminescent signal is then measured by the CL-1200i system to determine the presence or absence of HIV and/or HBV markers.

#### 2.2.1. HBV Analysis

Mindray CL-1200i uses a CLIA system for the quantitative determination of hepatitis B surface antigen HBsAg in human serum and its antibody to hepatitis Anti-HBs, but also for the qualitative determination of antigen HBeAg and its antibody Anti-HBe, and Anti-HBc. For quantitative determination, a two-site sandwich assay is used to determine the level of hepatitis B surface antigen; for qualitative determination, a competitive binding assay is used instead to determine the level of antibodies. CL-1200i system utilizes a paramagnetic microparticles platform with alkaline phosphatase (ALP)-labelled reagents and AMPPD substrates. 

The resulting chemiluminescent reaction is measured as relative light units (RLUs) by a photomultiplier built into the system. The amount of Anti-HBs present in the sample is proportional to the RLUs generated during the reaction. The Anti-HBs concentration can be determined via a calibration curve, which is built on an encoded Master Calibration Curve and three-level product calibrators. Instead, for qualitative determination, the presence or absence of analyte in the sample is determined by comparing the chemiluminescent signal in the reaction to the cut-off signal determined from system calibration. If the chemiluminescent signal is less than the cut-off signal (COI < 1.01), then the sample is considered reactive for the analytes; otherwise, if it is higher (COI ≥ 1.01), the sample is considered non-reactive for analyte.

Mindray CL-1200i supports continuous loading of samples, reagents, and consumables during the test; the operational capacity is confirmed around 120 tests/h.

#### 2.2.2. HIV Analysis

The CL-series HIV assay is a Chemiluminescent Immunoassay for the simultaneous qualitative determination of HIV p24 antigen and the antibodies to HIV-1 and/or HIV-2 in human serum or plasma. The assay is a two-site sandwich assay. In the first step, paramagnetic microparticles coated with mouse anti-HIV p24 mAb and microparticles coated with mouse specific HIV-1 and HIV-2 antigens are added into a reaction vessel. In the second step, ALP-labeled anti-HIV p24 monoclonal antibodies and ALP labeled HIV-1 and HIV-2 antigens are added to the reaction vessel to form a sandwich with microparticle-captured HIV p24 antigen, HIV-1, or HIV-2 antibodies, respectively. In the third step, the substrate is catalysed by ALP in the immune complex on the microparticles. The resulting chemiluminescent reaction is measured as relative light units (RLUs) by a photomultiplier. A direct relationship exists between the amount of p24 antigen, and the antibody to HIV-1 or HIV-2, or both in the sample and the RLUs generated during the reaction. The presence or absence of the HIV p24 antigen, HIV-1, or HIV-2 antibodies in the samples is determined by comparing the chemiluminescent signal in the reaction to the cut-off signal determined from calibration. Samples with a COI < 1.00 are considered negative for HIV and samples with a COI > 1 are considered positive.

### 2.3. Precision Study 

In order to assess the consistency and accuracy of the results obtained from the measurement process, a precision study was conducted. In particular, the precision of the CLIA system was evaluated by using the commercial negative and positive control materials (IQC) (Mindray Bio-Medical Electronics Co., Ltd., Shenzhen, China) recommended by the manufacturer for evaluating precision of HBsAg, Anti-HBs, HBeAg, Anti-HBe, Anti-HBc, Anti-HIV, HIVAb, and HIVAg. Precision estimation was performed by evaluating triplicate measurements of aliquots of the same samples, performed for a total of five non-consecutive days. The CL-series assay is designed to have a precision of ≤10% (within-device CV). The within-run and between-run were expressed as the coefficient of variations (CV%) calculated as the standard deviation divided by the mean value. Precision was determined by following National Committee for Clinical Laboratory Standards (NCCLS) Protocol EP5-A23. To estimate both intra-day precision and within-lab (total) precision repeatably, we used the software MetComp ver1.0 [13]. The manufacturers’ claims for precision were compared against precision results obtained for the IQC materials and verified by using the method suggested by EP15-A3 User Verification of Precision and Estimation of Bias (Approved Guideline—Third Edition. 2014. CLSI EP15-A3, Wayne, PA, USA). 

### 2.4. Linearity 

To determine whether the relationship between the input and output of the tests used followed a linear pattern or if there were any significant deviations from linearity, we performed linearity measurements. Linearity studies are important in evaluating the accuracy and reliability of measurement instruments or systems. By assessing linearity, we can understand the behavior of the system across a range of input values and identify potential nonlinearities that may affect the validity of the measurements. The linearity test was assessed using a series of mixes of three sample pools, prepared with different HBsAg values, using serial dilution (CLSI EP06). Three high-level sample pools were serially diluted with a low-level HBsAg antibody serum. The linearity for Anti-HBs was assessed analogously. All measurements were performed in triplicate. The linearity equation was calculated utilizing average value with standard deviation. The regression lines and their characteristics were calculated using Kaleidagraph software (version 3.5).

### 2.5. Carryover

Carryover refers to the residual effects or influence of a previous condition or event on subsequent conditions or measurements. It is commonly observed in scientific experiments, data analysis, and manufacturing processes. Carryover effects can introduce bias, distortion, or unwanted influence on results or outcomes. Addressing carryover effects is crucial in ensuring the validity and accuracy of experiments, analyses, and manufacturing processes. By acknowledging and properly managing carryover effects, we can minimize bias, make reliable inferences, and maintain consistent test quality. Carryover was performed for HBsAg, AntiHBs, and HIV Ag/Ab. The samples were divided in 11 “low” aliquots (L) and 10 “high” aliquots (H) and were loaded into the analyser in the following order: L, L, L, H, H, L, H, H, L, L, L, L, H, H, L, H, H, L, H, H, L [14]. The difference between the mean of the low measurements result after a high measurement, and the mean of the low measurements after a low measurement, is a measure for carry-over. The error limit is defined by 3 times the SD of the “low-low” results.

## 3. Results

### 3.1. Precision Study Results

For HBsAg, Anti-HBs, HBeAg, Anti-HBe, and Anti-HBc, the percent coefficient of variation values for the positive controls ranged from 3.4 % to 1.4% and 2.5% to 1%, for intra-assay and inter-assay, respectively. For the negative controls, the values ranged from 3.0% to 1.4% for between and from 1.2% to 0.9% for within run. The CV values obtained for all tests were almost all lower than those declared by the manufacturer, showing a performance better than that declared (Table 1). Only the value obtained for the HbeAg positive control was slightly higher.

The percent coefficient of variation value for the HIV negative control ranges from 3.1% for between run to 1.6% for within run. The positive HIV Ab control gave a CV 3.4% for between run and 3.0% for within run, while the CV HIV Ag positive control varied from 3.3% to 2.1%.

For the HIV test, the performance found was lower than that declared. For the HIVAb test, we found worse values for both the within run and the between run measurements. For HIVAg, the between run value agreed with the value declared, while for the within run, a CV value double of that declared by the manufacturer was found (Table 2). Nonetheless, this discrepancy is not statistically significant compared with the range of acceptability. 

### 3.2. Linearity Results

Linearity was demonstrated in the range of 0.05 IU/mL to 202.02 IU/mL for HBsAg, with three mixed low and high samples, respectively. The correlation coefficient R obtained was 0.99754, as shown in Figure 1.

The linearity test for Anti-HBs, obtained as previously described, was demonstrated in the range 0.02 IU/mL to 883.8 IU/mL. Data reported in the following Figure 2 show a regression coefficient R equivalent to 0.99626.

An important concern in laboratory medicine is to evaluate the possible carryover effect that can have a great impact on final results and can largely invalidate these kinds of studies. The clinical samples with the highest values and those with the lowest values (benchmark test), respectively, were tested in triplicate and the carryover effect was not observed for HBsAg, Anti- HBs, and HIV Ab/Ag (data not shown).

### 3.3. Comparison Study for HBV Test between Mindray and Abbott

The results obtained showed a good comparability between the Mindray CL-1200i and Abbott Architect (Table 3). The agreement between the results of the Abbott and Mindray CLIA ranged from 99% to 100% and the discrepancy rate from 0% to 1%. There were no false negative results, determining the sensitivity of each assay was 100% in our experiment. No interference or cross-reactivity was observed with known interfering substances and virologic markers. Only for the Anti-HbS test were two samples with false positives found, resulting in a positive agreement of 99%. Indeed, these two samples had values close to the cut-off.

### 3.4. Comparison Study for HIV Test between Mindray and Abbott

A total of 181 samples were measured using both assays. A comparison between the Mindray and the Abbott HIV test showed a good agreement (Table 3). All negative samples with the Abbott assay resulted negative with Mindray too. Two weakly positive samples with Abbott resulted negative with Mindray even after repeated tests. These samples were negative in the Western blot confirmation test and therefore considered negative samples. This resulted in a discrepancy rate of 0%.

## 4. Discussion

HBV and HIV infections are serious clinical problems due to their worldwide distribution and negative consequences [15]. HBV is one of the main causes of cirrhosis and hepatocellular carcinoma in Southeast Asia, China, and Africa [16]. According to UNAIDS data on the HIV and AIDS epidemic, it is estimated that, in 2020, out of the 37.7 million people living with the virus, there were 1.5 million new diagnoses. Among the 37.7 million people living with HIV infection, 36 million are adults and 1.7 million are children under the age of 15. Diagnostic tests for HIV and HBV are of great importance for the early diagnosis of the infection or for monitoring disease status and treatment response in HBV and HIV patients [17]. These tests are also crucial as screening tests for donated blood and blood components to assess the safety of blood samples for transfusion. Currently, diagnostics laboratories face an increasing demand for improved efficiency and response times. The recent development of CLIA assays offers advantages such as greater precision, reliability, technical simplicity, short delivery times, high-speed productivity, and full automation, especially for hospital laboratories with high work volumes [6,18]. The chemiluminescent immunoassay has numerous advantages, including a higher sensitivity than RIA and ELISA, the accurate detection of high concentrations, the absence of interfering emissions, the high stability of reagents and their conjugates, low reagents consumption, no stop reaction required, and a shorter incubation time than ELISA. Additionally, CLIAs provide improved specificity and a higher positive predictive value compared with conventional assays. Moreover, they are less expensive than the DNA tests used for the genomic quantification of HBV in serum. The Mindray CL-1220i HIV and HBV tests are based on a chemiluminescent reaction and offer the advantage of automated quantification and standardization. In the present study, we assessed the clinical performance of the Mindray CL-1200i compared with the Abbott Architect analytical system used in routine diagnostics. The percent agreement among the results ranged from 99% to 100%. Among the five hundred and forty-one samples tested by the two methods, discrepant results were obtained in two samples (1%). In our study, the results in routine clinical specimens were accurate overall, showing low variability and minimal discrepancy compared with the Abbott system. The lower limit of quantification was found to be close to the cut-off values obtained with the Abbott Architect system and adequate in efficiently quantifying all the serum samples tested in our study. The system showed a remarkably high sensitivity for detecting even minimal HBsAg levels. In general, the system offers the advantage of automated quantification, which reduces the potential for human error and ensures consistent and reliable results. The high sensitivity of the method allows for an high accuracy. This ensures accurate and precise results, contributing to the effectiveness of the diagnosis and monitoring of HBV and HIV infections. Concerning cost effectiveness, compared with DNA tests used for the genomic quantification of HBV in serum, the CL-1220i system is a less expensive option in line with other commercially available CLIA tests; this makes it economically feasible for diagnostic laboratories. On the other hand, limitations of the Mindray CL-1220i system may include dependency on a chemiluminescent reaction. The system’s reliance on chemiluminescent reactions for detection may limit its application in certain scenarios where other detection methods are more suitable or required. The clinical efficacy of the new, highly sensitive analyzer makes it suitable for the determination of HIV, HBsAg, Anti-HBs, HBeAg, Anti-Hbe, and Anti-HBc in routine laboratory analysis.

## 5. Conclusions

In conclusion, the data obtained suggest that the Mindray 1200i analyzer is suitable for the determination of HIV Antigen and Antibodies to Human Immunodeficiency Virus, as well as of HBsAg, Anti-HBs, HBeAg, Anti-Hbe, and Anti-HBc.

The CL-1200i, in particular, stands out as a robust and easy-to-use chemiluminescence analyzer, capable of processing a remarkable throughput of up to 180 tests per hour. Despite its compact size as a benchtop analyzer, the CL-1200i boasts a large capacity, accommodating up to 25 reagents and 60 samples. Moreover, it supports the continuous loading of samples, reagents, and consumables throughout the testing process. Additionally, the financial costs associated with each serum sample and maintenance expense are in line with other commercially available in vitro diagnostic tests, ensuring cost-effectiveness for laboratories. By seamlessly integrating trusted CLIA technology with the latest advancements in electrical and software enhancements, the CL-1200i enables easy operation and lightning-fast support. Its user-friendly interface and efficient design make it a valuable tool in routine analysis and an aid in the diagnosis of HIV-1/HIV-2 infections, as well as a reliable screening test for donated blood. Overall, the Mindray CL-1200i platform demonstrates high-level performance, consistently delivering accurate test results. Its capabilities and reliability make it a valuable asset when implemented in clinical laboratories for routine testing and diagnosing HIV infections. Moreover, it offers a good screening solution for donated blood, ensuring the safety and well-being of transfusion recipients.

## Figures and Tables

**Figure 1 diseases-11-00083-f001:**
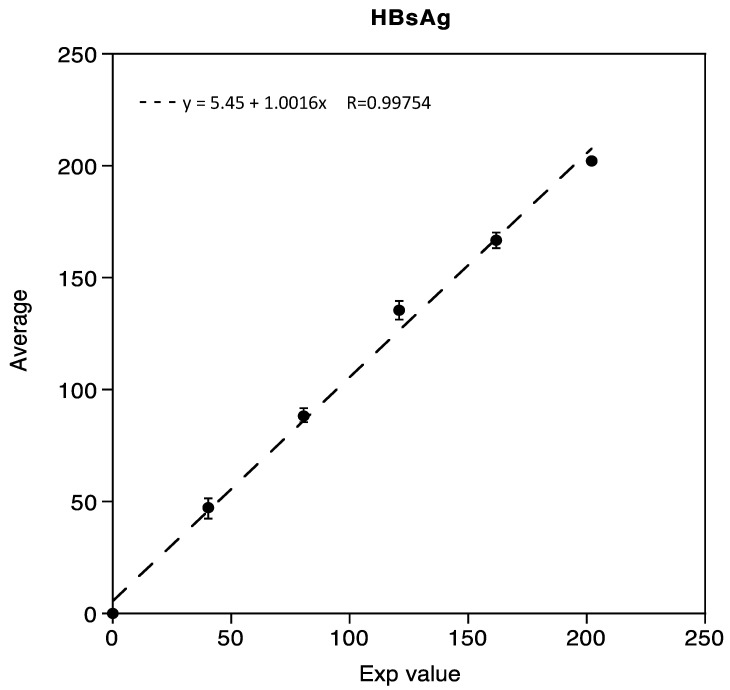
Linearity assessment of HBsAg immunoassay.

**Figure 2 diseases-11-00083-f002:**
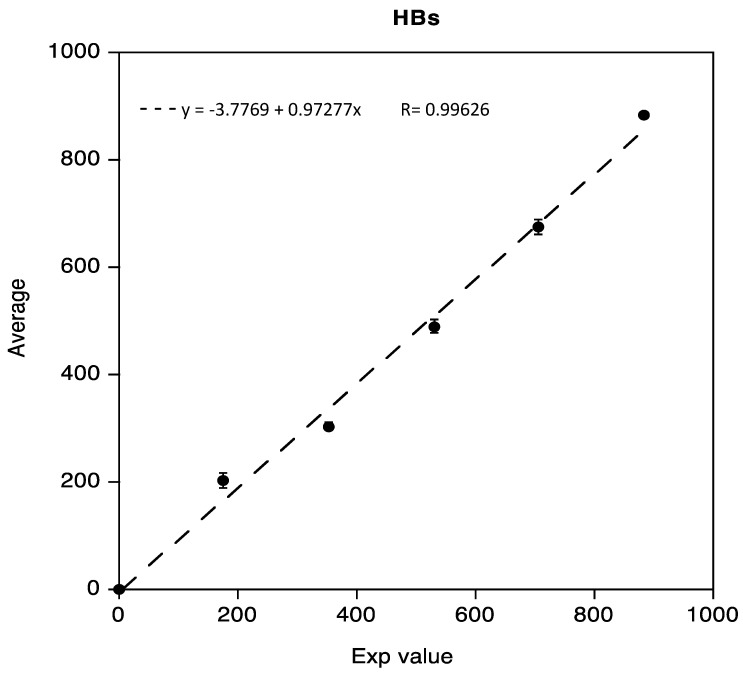
Linearity assessment of Anti-HBs immunoassay.

**Table 1 diseases-11-00083-t001:** Precision Study HBV.

		MINDRAY 1200i		MINDRAY 1200i	
		Between Run		Within Run	
TEST	CTRL	CV% Declared by the Manufacturer	CV% Laboratory	CV% Declared by the Manufacturer	CV% Laboratory
Anti HbC	N	3.6	1.8	2.9	1.1
Anti HbC	P	3.8	3.4	2.7	2.5
Anti HbS	N	3.3	1.8	3.1	0.9
Anti HbS	P	3.3	2.0	2.5	1.2
Anti HbE	N	2.6	1.7	3.3	1.1
Anti HbE	P	2.7	1.4	3.4	1.3
HbE Ag	N	5.3	1.4	2.1	1.2
HbE Ag	P	1.7	1.9	2.5	1.0
HbS Ag	N	7.2	3.0	3.5	0.9
HbS Ag	P	4.1	3.4	1.7	1.3

**Table 2 diseases-11-00083-t002:** Precision study HIV.

		MINDRAY 1200i		MINDRAY 1200i	
		Within Run		Between Run	
TEST CTRL		CV% Declared by the Manufacturer	CV% Laboratory	CV% Declared by the Manufacturer	CV% Laboratory
AntiHIV	N	2.4	1.6	3.4	3.1
HIV Ab	P	1.2	3.0	3.0	3.4
HIV Ag	P	1.2	2.1	3.7	3.3

**Table 3 diseases-11-00083-t003:** Comparison of HIV and HBV between Mindray and Abbott.

HIV							
	Sample No.	Positive for Abbott	Negative for Abbott	Negative Agreement	Positive Agreement	Discrepancy Sample No.	Discrepancy Rate
HIV	181	34	147	100%	100%	0	0%
HBV							
HbsAg	201	51	150	100%	100%	0	0%
Anti-Hbs	155	134	21	100%	99%	2	1%
HbeAg	54	10	44	100%	100%	0	0%
Anti Hbe	50	15	35	100%	100%	0	0%
Anti Hbc	81	18	63	100%	100%	0	0%

## Data Availability

The data presented in this study are available on request from the corresponding author.

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
