# Peer review of "Performance Evaluation of the New Chemiluminescence Immunoassay CL-1200i for HBV, HIV Panels"

_diseases, 2023, doi:10.3390/diseases11020083_

Round 1

Reviewer 1 Report

Title: Performance evaluation of the new Chemiluminescence Immunoassay CL-1200i HBV, HIV panels.

Manuscript ID: 2393802

I recommended that manuscript could be accepted with MAJOR MODIFICATIONS.

GENERAL COMMENTS

-          In addition, carefully review: (i) abbreviations: define the first time that is mentioned and use in following text, (ii) typographic mistakes, and (iii) English language and grammar.

ABSTRACT

-          Include quantitative results.

INTRODUCTION

-          Ideas or paragraphs should be connected and condensed the information. Specify if previous results of with Mindray 1200i for HBV- and HIV infection detection have been documented in scientific literature.

MATERIAL AND METHODS

-          Include period of sample collection.

-          Were the collected 541/181 sample representative from total of received sample? In the study, did the author apply some criterions of inclusion / exclusion?  

-          Specify program (with version or year) to carry out the analysis of variance (ANOVA), and signification value.

-          Include version or year of Kaleidagraph software.

RESULTS

-          In the text, in each comparison, “p” value should be inserted.

-          Include statistical analysis in tables 1 and 2 (between manufacturer and laboratory).

DISCUSION

-          This chapter should be re-writing, due to: (i) complete ideas in a paragraph instead sentences, (ii) advantages and limitation of Mindray CL-1220i system could be mentioned, and (iii) increase discussion related with results and recommendation of its implementation.

English language could be review.

Author Response

GENERAL COMMENTS

-          In addition, carefully review: (i) abbreviations: define the first time that is mentioned and use in following text, (ii) typographic mistakes, and (iii) English language and grammar.

As suggested by the reviewer the English language and grammar, abbreviations and typographic mistakes, have been revised.

ABSTRACT

-          Include quantitative results.

A sentence has been added in the abstract.

INTRODUCTION

-          Ideas or paragraphs should be connected and condensed the information. Specify if previous results of with Mindray 1200i for HBV- and HIV infection detection have been documented in scientific literature.

As suggested by the reviewer, we revised the introduction section.

To our knowledge no other scientific literature reported HBV and HIV detection data with Mindray 1200i platform.

 MATERIAL AND METHODS

-          Include period of sample collection. 

The period of collection has been introduced in paragraph 2.1

-          Were the collected 541/181 sample representative from total of received sample? In the study, did the author apply some criterions of inclusion / exclusion?  

The collected samples are representative from total of the received samples, no inclusion/exclusion criteria have been applied.

-          Specify program (with version or year) to carry out the analysis of variance (ANOVA), and signification value.

The use of ANOVA for analysis of variance reported was a typo error. The software used is MetComp ver1.0 described in the SIBIOC document Biochimica Clinica, May 2016, Vivaldi et al.( DOI: 10.19186/BC_2016.006). The specific software returns ACCEPTABLE or NOT ACCEPTABLE accordingly with p value calculated, without showing the value.

Material and Methods section has been modified reporting the reference mentioned above.

-          Include version or year of Kaleidagraph software.

The version of the software has been reported in paragraph 2.5

RESULTS

-          In the text, in each comparison, “p” value should be inserted. 

-          Include statistical analysis in tables 1 and 2 (between manufacturer and laboratory).

The use of ANOVA for analysis of variance reported, was a typo error. The software used is MetComp ver1.0 described in the SIBIOC document Biochimica Clinica, May 2016, Vivaldi et al.(DOI: 10.19186/BC_2016.006). The test returns ACCEPTABLE or NOT ACCEPTABLE and give no p values.

 DISCUSION

-          This chapter should be re-writing, due to: (i) complete ideas in a paragraph instead sentences, (ii) advantages and limitation of Mindray CL-1220i system could be mentioned, and (iii) increase discussion related with results and recommendation of its implementation.

The paragraph has been revised according to reviewer suggestions and a paragraph has been added.

Reviewer 2 Report

The aim of this study is to compare the performance of the Abbott system and of the new analyser Mindray 1200i in the detection of HBV- and HIV infection, including Anti HbC, Anti HbS, Anti HbE, HbE Ag, HbS Ag, HIV p24 antigen and the antibodies to HIV-1 and/or HIV-2. Some revisions are suggested:

1) From the “Introduction” part, it is not clear why do the authors simultaneously detect HIV and HBV. Are they just two independent models for the diagnosis of infection diseases? The significance should be highlighted.

2) Too many paragraphs are present in section “Introduction” and “Discussion”.

3) Table 1, the relevant raw data should also be provided in a scatter plot to give direct view.

4) Line 214, the authors claim that “… this discrepancy is not statistically significant…”. Can they show the comparison by raw data?

5) Section 3.3 and 3.4, the raw data should also be provided.

Author Response

Some revisions are suggested:

  • From the “Introduction” part, it is not clear why do the authors simultaneously detect HIV and HBV. Are they just two independent models for the diagnosis of infection diseases? The significance should be highlighted.

Worldwide, hepatitis B virus infection, is one of the most common coinfections with human immunodeficiency virus and has become a major threat to the survaival of HIV-infected persons. In line with this clinical requirement, the aim of this study is to compare the performance of the Abbott system and of the new Mindray 1200i analyser in the detecting HBV and HIV infections.

The sentences have been also added in the introduction.

  • Too many paragraphs are present in section “Introduction” and “Discussion”.

The paragraphs have been reduced.

3) Table 1, the relevant raw data should also be provided in a scatter plot to give direct view.  

Indeed, according to us the representation using tables is clearer than the graph.

4) Line 214, the authors claim that “… this discrepancy is not statistically significant…”. Can they show the comparison by raw data? 

Precision for laboratory has been calculated trough a 3x5 protocol, as described in material and method, and compared with the CV value declared by manufacturer using the software MetComp ver1.0 described in the SIBIOC document Biochimica Clinica, May 2016, Vivaldi et al.(DOI: 10.19186/BC_2016.006).

5) Section 3.3 and 3.4, the raw data should also be provided.

A table has been added with the raw data (table 3)

Reviewer 3 Report

Title should be changed, I think better is ... for HBV and HIV serum panels.

the word "highlight"" on line 51, better to use ""detect""

In analysis HIV-1 and HIV-2 are mentioned, but later in Table 2 only HIV is presented. Can this be explained in more detail?

line 228: ... ""to try out..."" better change in ""to evaluate""

The authors want to promote the Mindray CL-1200i equipment as important extra tool for evaluation of serum samples, perhaps also financial costs or costs per serum sample or maintenance costs can be discussed?

Some small editing and proof reading would be recommended.

Author Response

-Title should be changed, I think better is ... for HBV and HIV serum panels.

Title has been modified according to reviewer suggestion.

  • The word "highlight"" on line 51, better to use ""detect""

The word has been corrected

  • In analysis HIV-1 and HIV-2 are mentioned, but later in Table 2 only HIV is presented. Can this be explained in more detail?

The test is directed to p24 antigen that belong both to HIV-1 both to HIV-2. Indeed in case of positive result it means that one or both the antibodies are present in the sample.

  • line 228: ... ""to try out..."" better change in ""to evaluate""

The expression has been changed accordingly to reviewer suggestion.

  • The authors want to promote the Mindray CL-1200i equipment as important extra tool for evaluation of serum samples, perhaps also financial costs or costs per serum sample or maintenance costs can be discussed?

Both maintainance and test are in line with other commercially available in vitro diagnostic test (a sentence has been added in the conclusion paragraph)

Round 2

Reviewer 1 Report

The manuscript has been improved and could be accepted in present form.

Reviewer 2 Report

The authors have improved the manuscript. I've no further comments now.